# A Screening Rule for $\ell_1$-Regularized Ising Model Estimation

**Zhaobin Kuang[1], Sinong Geng[2], David Page[3]**
University of Wisconsin
zkuang@wisc.edu[1], sgeng2@wisc.edu[2], page@biostat.wisc.edu[3]

## Abstract

We discover a screening rule for $\ell_1$-regularized Ising model estimation. The simple closed-form screening rule is a necessary and sufficient condition for exactly recovering the blockwise structure of a solution under any given regularization parameters. With enough sparsity, the screening rule can be combined with various optimization procedures to deliver solutions efficiently in practice. The screening rule is especially suitable for large-scale exploratory data analysis, where the number of variables in the dataset can be thousands while we are only interested in the relationship among a handful of variables within moderate-size clusters for interpretability. Experimental results on various datasets demonstrate the efficiency and insights gained from the introduction of the screening rule.

## 1 Introduction

While the field of statistical learning with sparsity [Hastie et al., 2015] has been steadily rising to prominence ever since the introduction of the lasso (least absolute shrinkage and selection operator) at the end of the last century [Tibshirani, 1996], it was not until the recent decade that various *screening rules* debuted to further equip the ever-evolving optimization arsenals for some of the most fundamental problems in sparse learning such as $\ell_1$-regularized generalized linear models (GLMs, Friedman et al. 2010) and inverse covariance matrix estimation [Friedman et al., 2008]. Screening rules, usually in the form of an analytic formula or an optimization procedure that is extremely fast to solve, can accelerate learning drastically by leveraging the inherent sparsity of many high-dimensional problems. Generally speaking, screening rules can identify a significant portion of the zero components of an optimal solution beforehand at the cost of minimal computational overhead, and hence substantially reduce the dimension of the parameterization, which makes possible efficient computation for large-scale sparse learning problems.

Pioneered by Ghaoui et al. 2010, various screening rules have emerged to speed up learning for generative models (e.g. Gaussian graphical models) as well as for discriminative models (e.g. GLMs), and for continuous variables (e.g. lasso) as well as for discrete variables (e.g. logistic regression, support vector machines). Table 1 summarizes some of the iconic work in the literature, where, to the best of our knowledge, screening rules for generative models with discrete variables are still notably absent.

Contrasted with this notable absence is the ever stronger craving in the big data era for scaling up the learning of generative models with discrete variables, especially in a blockwise structure identification setting. For example, in gene mutation analysis [Wan et al., 2015, 2016], among tens of thousands of sparse binary variables representing mutations of genes, we are interested in identifying a handful of mutated genes that are connected into various blocks and exert synergistic effects on the cancer. While a sparse Ising model is a desirable choice, for such an application the scalability of the model could fail due to the innate $\mathcal{NP}$-hardness [Karger and Srebro, 2001] of inference, and hence maximum likelihood learning, owing to the partition function. To date, even with modern

Table 1: Screening rules in the literature at a glance

| | Discriminative Models | Generative Models |
|---|---|---|
| Continuous Variables | Ghaoui et al. 2010, Tibshirani et al. 2012 Liu et al. 2013, Wang et al. 2013, Fercoq et al. 2015, Xiang et al. 2016, Lee et al. 2017 | Banerjee et al. 2008 Honorio and Samaras 2010 Witten et al. 2011,Mazumder and Hastie 2012 Danaher et al. 2014, Luo et al. 2014 Yang et al. 2015 |
| Discrete Variables | Ghaoui et al. 2010, Tibshirani et al. 2012 Wang et al. 2014, Ndiaye et al. 2015 | **?** |

approximation techniques, a typical application with sparse discrete graphical models usually involves only hundreds of variables [Viallon et al., 2014, Barber et al., 2015, Vuffray et al., 2016].

Between the need for the scalability of high-dimensional Ising models and the absence of screening rules that are deemed crucial to accelerated and scalable learning, we have a technical gap to bridge: *can we identify screening rules that can speed up the learning of $\ell_1$-regularized Ising models?* The major contribution of this paper is to give an affirmative answer to this question. Specifically, we show the following.

- The screening rule is a simple closed-form formula that is a necessary and sufficient condition for exact blockwise structure recovery of the solution with a given regularization parameter. Upon the identification of blockwise structures, different blocks of variables can be considered as different Ising models and can be solved separately. The various blocks can even be solved *in parallel* to attain further efficiency. Empirical results on both simulated and real-world datasets demonstrate the tremendous efficiency, scalability, and insights gained from the introduction of the screening rule. Efficient learning of $\ell_1$-regularized Ising models from thousands of variables on a single machine is hence readily attainable.

- As an initial attempt to fill in the vacancy illustrated in Table 1, our work is instructive to further exploration of screening rules for other graphical models with discrete random variables, and to combining screening rules with various optimization methods to facilitate better learning. Furthermore, compared with its Gaussian counterpart, where screening rules are available (Table 1) and learning is scalable [Hsieh et al., 2013], the proposed screening rule is especially valuable and desperately needed to address the more challenging learning problem of sparse Ising models.

We defer all the proofs in the paper to the supplement and focus on providing intuition and interpretation of the technical results in the paper.

## 2 Notation and Background

### 2.1 Ising Models

Let $X = [X_1, X_2, \cdots, X_p]^\top$ be a $p \times 1$ binary random vector, with $X_i \in \{-1, 1\}$, and $i \in \{1, 2, \cdots, p\} \triangleq V$. Let there be a dataset $\mathbb{X}$ with $n$ independent and identically distributed samples of $X$, denoted as $\mathbb{X} = \{x^{(1)}, x^{(2)}, \cdots, x^{(n)}\}$. Here, $x^{(k)}$ is a $p \times 1$ vector of assignments that realizes $X$, where $k \in \{1, 2, \cdots, n\}$. We further use $x_i^{(k)}$ to denote the $i^{th}$ component of the $k^{th}$ sample in the dataset. Let $\theta \in \Theta$ be a $p \times p$ *symmetric* matrix whose diagonal entries are zeros. An Ising model [Wan et al., 2016] with the parameterization $\theta$ is:

$$\mathrm{P}_\theta(x) = \frac{1}{Z(\theta)} \exp\left(\sum_{i=1}^{p-1} \sum_{j>i}^{p} \theta_{ij} x_i x_j\right), \tag{1}$$

where $\theta_{ij}$ represents the component of $\theta$ at the $i^{th}$ row and the $j^{th}$ column, and $x_i$ and $x_j$ represent the $i^{th}$ and the $j^{th}$ components of $x$, respectively. $Z(\theta)$ is a normalization constant, partition function, that ensures the probabilities sum up to one. The partition function is given as $Z(\theta) = \sum_{x \in \{-1,1\}^p} \exp\left(\sum_{i=1}^{p-1} \sum_{j>i}^{p} \theta_{ij} x_i x_j\right)$. Note that for ease of presentation, we consider Ising models with only pairwise interaction/potential here. Generalization to Ising models with unary potentials is given in Section 6.

## 2.2 Graphical Interpretation

With the notion of the probability given by an Ising model in (1), estimating an $\ell_1$-regularized Ising model is defined as finding $\hat{\theta}$, the penalized maximum likelihood estimator (MLE) under the lasso penalty:

$$
\begin{aligned}
\hat{\theta} &= \arg\max_{\theta} \frac{1}{n} \sum_{k=1}^{n} \log \mathrm{P}_{\theta}\left(x^{(k)}\right) - \frac{\lambda}{2} \|\theta\|_1 \\
&= \arg\min_{\theta} -\frac{1}{n} \sum_{k=1}^{n} \sum_{i=1}^{p-1} \sum_{j>i}^{p} \theta_{ij} x_i^{(k)} x_j^{(k)} + A(\theta) + \frac{\lambda}{2} \|\theta\|_1.
\end{aligned}
\tag{2}
$$

Here, $A(\theta) = \log Z(\theta)$ is the log-partition function; $\|\theta\|_1 = \sum_{i=1}^{p} \sum_{j=1}^{p} |\theta_{ij}|$ is the lasso penalty that encourages a sparse parameterization. $\lambda \geq 0$ is a given regularization parameter. Using $\frac{\lambda}{2}$ is suggestive of the symmetry of $\theta$ so that $\frac{\lambda}{2} \|\theta\|_1 = \lambda \sum_{i=1}^{p-1} \sum_{j>i}^{p} |\theta_{ij}|$, which echoes the summations in the negative log-likelihood function. Note that $\theta$ corresponds to the adjacency matrix constructed by the $p$ components of $X$ as nodes, and $\theta_{ij} \neq 0$ indicates that there is an edge between $X_i$ and $X_j$. We further denote a *partition* of $V$ into $L$ blocks as $\{C_1, C_2, \cdots, C_L\}$, where $C_l, C_{l'} \subseteq V$, $C_l \cap C_{l'} = \emptyset$, $\bigcup_{l=1}^{L} C_l = V$, $l \neq l'$, and for all $l, l' \in \{1, 2, \cdots, L\}$. Without loss of generality, we assume that the nodes in different blocks are ordered such that if $i \in C_l$, $j \in C_{l'}$, and $l < l'$, then $i < j$.

## 2.3 Blockwise Solutions

We introduce the definition of a blockwise parameterization:

**Definition 1.** We call $\theta$ *blockwise* with respect to the partition $\{C_1, C_2, \cdots, C_L\}$ if $\forall l$ and $l' \in \{1, 2, \cdots, L\}$, where $l \neq l'$, and $\forall i \in C_l, \forall j \in C_{l'}$, we have $\theta_{ij} = 0$.

When $\theta$ is blockwise, we can represent $\theta$ in a block diagonal fashion:

$$
\theta = \mathrm{diag}\left(\theta_1, \theta_2, \cdots, \theta_L\right),
\tag{3}
$$

where $\theta_1, \theta_2, \cdots,$ and $\theta_L$ are symmetric matrices that correspond to $C_1, C_2, \cdots,$ and $C_L$, respectively. Note that if we can identify the blockwise structure of $\hat{\theta}$ in advance, we can solve each block independently (See A.1). Since the size of each block could be much smaller than the size of the original problem, each block could be much easier to learn compared with the original problem. Therefore, efficient identification of blockwise structure could lead to substantial speedup in learning.

# 3 The Screening Rule

## 3.1 Main Results

The preparation in Section 2 leads to the discovery of the following strikingly simple screening rule presented in Theorem 1.

**Theorem 1.** Let a partition of V, $\{C_1, C_2, \cdots, C_L\}$, be given. Let the dataset $\mathbb{X} = \{x^{(1)}, x^{(2)}, \cdots, x^{(n)}\}$ be given. Define $\mathbb{E}_{\mathbb{X}} X_i X_j = \frac{1}{n} \sum_{k=1}^{n} x_i^{(k)} x_j^{(k)}$. A necessary and sufficient condition for $\hat{\theta}$ to be blockwise with respect to the given partition is that

$$
|\mathbb{E}_{\mathbb{X}} X_i X_j| \leq \lambda,
\tag{4}
$$

for all $l$ and $l' \in \{1, 2, \cdots, L\}$, where $l \neq l'$, and for all $i \in C_l, j \in C_{l'}$.

In terms of exact blockwise structure identification, Theorem 1 provides a foolproof (necessary and sufficient) and yet easily checkable result by comparing the absolute second empirical moments $|\mathbb{E}_{\mathbb{X}} X_i X_j|$'s with the regularization parameter $\lambda$. We also notice the remarkable similarity between the proposed screening rule and the screening rule for Gaussian graphical model blockwise structure identification in Witten et al. 2011, Mazumder and Hastie 2012. In the Gaussian case, the screening rule can be attained by simply replacing the second empirical moment matrix in (4) with the sample

---

**Algorithm 1** Blockwise Minimization

---

1: **Input:** dataset $\mathbb{X}$, regularization parameter $\lambda$.
2: **Output:** $\hat{\theta}$.
3: $\forall i, j \in V$ such that $j > i$, compute the second empirical moments $\mathbb{E}_{\mathbb{X}} X_i X_j$'s .
4: Identify the partition $\{C_1, C_2, \cdots, C_L\}$ using the second empirical moments from the previous step and according to Witten et al. [2011], Mazumder and Hastie [2012].
5: $\forall l \in L$, perform blockwise optimization over $C_l$ for $\hat{\theta}_l$.
6: Ensemble $\hat{\theta}_l$'s according to (3) for $\hat{\theta}$.
7: **Return** $\hat{\theta}$.

---

covariance matrix. While the exact solution in the Gaussian case can be computed in polynomial time, estimating an Ising model via maximum likelihood in general is $\mathcal{NP}$-hard . However, as a consequence of applying the screening rule, the blockwise structure of an $\ell_1$-regularized Ising model can be determined *as easily as* the blockwise structure of a Gaussian graphical model, despite the fact that within each block, exact learning of a sparse Ising model could still be challenging.

Furthermore, the screening rule also provides us a principal approach to leverage sparsity for the gain of efficiency: by increasing $\lambda$, the nodes of the Ising model will be shattered into smaller and smaller blocks, according to the screening rule. Solving many Ising models with small blocks of variables is amenable to both estimation algorithm and parallelism.

### 3.2 Regularization Parameters

The screening rule also leads to a significant implication to the range of regularization parameters in which $\hat{\theta} \neq 0$. Specifically, we have the following theorem.

**Theorem 2.** Let the dataset $\mathbb{X} = \{x^{(1)}, x^{(2)}, \cdots, x^{(n)}\}$ be given, and let $\lambda = \lambda_{\max}$ represent the smallest regularization parameter such that $\hat{\theta} = 0$ in (2). Then $\lambda_{\max} = \max_{i,j \in V, i \neq j} |\mathbb{E}_{\mathbb{X}} X_i X_j| \leq 1$.

With $\lambda_{\max}$, one can decide the range of regularization parameters, $[0, \lambda_{\max}]$, that generates graphs with nonempty edge sets, which is an important first step for pathwise optimization algorithms (a.k.a. homotopy algorithms) that learn the solutions to the problem under a range of $\lambda$'s. Furthermore, the fact that $\lambda_{\max} \leq 1$ for any given dataset $\mathbb{X}$ suggests that comparison across different networks generated by different datasets is comprehensible. Finally, in Section 4, $\lambda_{\max}$ will also help to establish the connection between the screening rule for exact learning and some of the popular inexact (alternative) learning algorithms in the literature.

### 3.3 Fully Disconnected Nodes

Another consequence of the screening rule is the necessary and sufficient condition that determines the regularization parameter with which a node is fully disconnected from the remaining nodes:

**Corollary 1.** Let the dataset $\mathbb{X} = \{x^{(1)}, x^{(2)}, \cdots, x^{(n)}\}$ be given. $X_i$ is fully disconnected from the remaining nodes in $\hat{\theta}$, where $i \in V$ (i.e., $\hat{\theta}_{ij} = \hat{\theta}_{ji} = 0$, $\forall j \in V \setminus \{i\}$), if and only if $\lambda \geq \max_{j \in V \setminus \{i\}} |\mathbb{E}_{\mathbb{X}} X_i X_j|$.

In high-dimensional exploratory data analysis, it is usually the case that *most* of the variables are fully disconnected [Danaher et al., 2014, Wan et al., 2016]. In this scenario, Corollary 1 provides a regularization parameter threshold with which we can identify *exactly* the subset of fully disconnected nodes. Since we can choose a threshold large enough to make *any* nodes fully disconnected, we can discard a significant portion of the variables efficiently and flexibly at will with exact optimization guarantees due to Corollary 1. By discarding the large portion of fully disconnected variables, the learning algorithm can focus on only a moderate number of connected variables, which potentially results in a substantial efficiency gain.

### 3.4 Blockwise Minimization

We conclude this section by providing the blockwise minimization algorithm in Algorithm 1 due to the screening rule. Note that both the second empirical moments and the partition of $V$ in the

algorithm can be computed in $O(p^2)$ operations [Witten et al., 2011, Mazumder and Hastie, 2012]. On the contrary, the complexity of the exact optimization of a block of variables grows exponentially with respect to the maximal clique size of that block. Therefore, by encouraging enough sparsity, the blockwise minimization due to the screening rule can provide remarkable speedup by not only shrinking the size of the blocks in general but also potentially reducing the size of cliques within each block via eliminating enough edges.

## 4  Applications to Inexact (Alternative) Methods

We now discuss the interplay between the screening rule and two popular inexact (alternative) estimation methods: node-wise (NW) logistic regression [Wainwright et al., 2006, Ravikumar et al., 2010] and the pseudolikelihood (PL) method [Höfling and Tibshirani, 2009]. In what follows, we use $\hat{\theta}^{\text{NW}}$ and $\hat{\theta}^{\text{PL}}$ to denote the solutions given by the node-wise logistic regression method and the pseudolikelihood method, respectively. NW can be considered as an *asymmetric* pseudolikelihood method (i.e., $\exists i,j \in V$ such that $i \neq j$ and $\hat{\theta}^{\text{NW}}_{ij} \neq \hat{\theta}^{\text{NW}}_{ji}$), while PL is a pseudolikelihood method that is similar to NW but imposes additional *symmetric* constraints on the parameterization (i.e., $\forall i,j \in V$ where $i \neq j$, we have $\hat{\theta}^{\text{PL}}_{ij} = \hat{\theta}^{\text{PL}}_{ji}$).

Our incorporation of the screening rule to the inexact methods is straightforward: after using the screening rule to identify different blocks in the solution, we use inexact methods to solve each block for the solution. As shown in Section 3, when combined with exact optimization, the screening rule is foolproof for blockwise structure identification. However, in general, when combined with inexact methods, the proposed screening rule is not foolproof any more because the screening rule is derived from the exact problem in (2) instead of the approximate problems such as NW and PL. We provide a toy example in A.6 to illustrate mistakes made by the screening rule when combined with inexact methods. Nonetheless, as we will show in this section, NW and PL are deeply connected to the screening rule, and when given a large enough regularization parameter, the application of the screening rule to NW and PL can be lossless in practice (see Section 5). Therefore, when applied to NW and PL, the proposed screening rule can be considered as a strong rule (i.e., a rule that is not foolproof but barely makes mistakes) and an optimal solution can be safeguarded by adjusting the screened solution to optimality based on the KKT conditions of the inexact problem [Tibshirani et al., 2012].

### 4.1  Node-wise (NW) Logistic Regression and the Pseudolikelihood (PL) Method

In NW, for each $i \in V$, we consider the conditional probability of $X_i$ upon $X_{\setminus i}$, where $X_{\setminus i} = \{X_t \mid t \in V \setminus \{i\}\}$. This is equivalent to solving $p$ $\ell_1$-regularized logistic regression problems separately, i.e., $\forall i \in V$ :

$$\hat{\theta}^{\text{NW}}_{\setminus i} = \arg\min_{\theta_{\setminus i}} \frac{1}{n} \sum_{k=1}^{n} \left[ -y_i^{(k)} \eta_{\setminus i}^{(k)} + \log\left(1 + \exp\left(\eta_{\setminus i}^{(k)}\right)\right) \right] + \lambda \left\| \theta_{\setminus i} \right\|_1, \tag{5}$$

where $\eta_{\setminus i}^{(k)} = \theta_{\setminus i}^{\top}(2x_{\setminus i}^{(k)})$, $y_i^{(k)} = 1$ represents a successful event $x_i^{(k)} = 1$, $y_i^{(k)} = 0$ represents an unsuccessful event $x_i^{(k)} = -1$, and

$$\theta_{\setminus i} = \begin{bmatrix} \theta_{i1} & \theta_{i2} & \cdots & \theta_{i(i-1)} & \theta_{i(i+1)} & \cdots & \theta_{ip} \end{bmatrix}^{\top},$$
$$x_{\setminus i}^{(k)} = \begin{bmatrix} x_{i1}^{(k)} & x_{i2}^{(k)} & \cdots & x_{i(i-1)}^{(k)} & x_{i(i+1)}^{(k)} & \cdots & x_{ip}^{(k)} \end{bmatrix}^{\top}.$$

Note that $\hat{\theta}^{\text{NW}}$ constructed from $\hat{\theta}^{\text{NW}}_{\setminus i}$'s is asymmetric, and ad hoc post processing techniques are used to generate a symmetric estimation such as setting each pair of elements from $\hat{\theta}^{\text{NW}}$ in symmetric positions to the one with a larger (or smaller) absolute value.

On the other hand, PL can be considered as solving all $p$ $\ell_1$-regularized logistic regression problems in (5) jointly with symmetric constraints over the parameterization [Geng et al., 2017]:

$$\hat{\theta}^{\text{PL}} = \arg\min_{\theta \in \Theta} \frac{1}{n} \sum_{k=1}^{n} \sum_{i=1}^{p} \left[ -y_i^{(k)} \xi_i^{(k)} + \log\left(1 + \exp\left(\xi_i^{(k)}\right)\right) \right] + \frac{\lambda}{2} \left\| \theta \right\|_1, \tag{6}$$

where $\xi_i^{(k)} = \sum_{j \in V \setminus \{i\}} 2\theta_{\min\{i,j\},\max\{i,j\}} x_j^{(k)}$. That is to say, if $i < j$, then $\theta_{\min\{i,j\},\max\{i,j\}} = \theta_{ij}$; if $i > j$, then $\theta_{\min\{i,j\},\max\{i,j\}} = \theta_{ji}$. Recall that $\Theta$ in (6) defined in Section 2.1 represents a space of symmetric matrices whose diagonal entries are zeros.

## 4.2 Regularization Parameters in NW and PL

Since the blockwise structure of a solution is given by the screening rule under a *fixed* regularization parameter, the ranges of regularization parameters under which NW and PL can return nonzero solutions need to be linked to the range $[0, \lambda_{\max}]$ in the exact problem. Theorem 3 and Theorem 4 establish such relationships for NW and PL, respectively.

**Theorem 3.** Let the dataset $\mathbb{X} = \{x^{(1)}, x^{(2)}, \cdots, x^{(n)}\}$ be given, and let $\lambda = \lambda_{\max}^{\mathrm{NW}}$ represent the smallest regularization parameter such that $\hat{\theta}_{\setminus i}^{\mathrm{NW}} = 0$ in (5), $\forall i \in V$. Then $\lambda_{\max}^{\mathrm{NW}} = \lambda_{\max}$.

**Theorem 4.** Let the dataset $\mathbb{X} = \{x^{(1)}, x^{(2)}, \cdots, x^{(n)}\}$ be given, and let $\lambda = \lambda_{\max}^{\mathrm{PL}}$ represent the smallest regularization parameter such that $\hat{\theta}^{\mathrm{PL}} = 0$ in (6), then $\lambda_{\max}^{\mathrm{PL}} = 2\lambda_{\max}$.

Let $\lambda$ be the regularization parameter used in the exact problem. A strategy is to set the corresponding $\lambda^{\mathrm{NW}} = \lambda$ when using NW and $\lambda^{\mathrm{PL}} = 2\lambda$ when using PL, based on the range of regularization parameters given in Theorem 3 and Theorem 4 for NW and PL. Since the magnitude of the regularization parameter is suggestive of the magnitude of the gradient of the unregulated objective, the proposed strategy leverages that the magnitudes of the gradients of the unregulated objectives for NW and PL are roughly the same as, and roughly twice as large as, that of the unregulated exact objective, respectively.

This observation has been made in the literature of binary pairwise Markov networks [Höfling and Tibshirani, 2009, Viallon et al., 2014]. Here, by Theorem 3 and Theorem 4, we demonstrate that this relationship is exactly true if the optimal parameterization is zero. Höfling and Tibshirani 2009 even further exploits this observation in PL for exact optimization. Their procedure can be viewed as iteratively solving adjusted PL problems regularized by $\lambda^{\mathrm{PL}} = 2\lambda$ in order to obtain an exact solution regularized by $\lambda$. The close quantitative correspondence between the derivatives of the inexact objectives and that of the exact objective also provides insights into why combing the screening rule with inexact methods does not lose much in practice.

## 4.3 Preservation for Fully Disconnectedness

While the screening rule is not foolproof when combined with NW and PL, it turns out that in terms of identifying fully disconnected nodes, the necessary and sufficient condition in Corollary 1 can be preserved when applying NW with caution, as shown in the following.

**Theorem 5.** Let the dataset $\mathbb{X} = \{x^{(1)}, x^{(2)}, \cdots, x^{(n)}\}$ be given. Let $\hat{\theta}_{\min}^{\mathrm{NW}} \in \Theta$ denote a symmetric matrix derived from $\hat{\theta}^{\mathrm{NW}}$ by setting each pair of elements from $\hat{\theta}^{\mathrm{NW}}$ in symmetric positions to the one with a smaller absolute value. A sufficient condition for $X_i$ to be fully disconnected from the remaining nodes in $\hat{\theta}_{\min}^{\mathrm{NW}}$, where $i \in V$, is that $\lambda^{\mathrm{NW}} \geq \max_{j \in V \setminus \{i\}} |\mathbb{E}_{\mathbb{X}} X_i X_j|$. Furthermore, when $\hat{\theta}_{\setminus i}^{\mathrm{NW}} = 0$, the sufficient condition is also necessary.

In practice, the utility of Theorem 5 is to provide us a lower bound for $\lambda$ above which we can fully disconnect $X_i$ (sufficiency). Moreover, if $\hat{\theta}_{\setminus i}^{\mathrm{NW}} = 0$ also happens to be true, which is easily verifiable, we can conclude that such a lower bound is tight (necessity).

## 5 Experiments

Experiments are conducted on both synthetic data and real world data. We will focus on *efficiency* in Section 5.1 and discuss *support recovery* performance in Section 5.2. We consider three synthetic networks (Table 2) with 20, 35, and 50 blocks of 20-node, 35-node, and 50-node subnetworks, respectively. To demonstrate the estimation of networks with unbalanced-size subnetworks, we also consider a 46-block network with power law degree distributed subnetworks of sizes ranging from 5 to 50. Within each network, the subnetwork is generated according to a power law degree distribution, which mimics the structure of a biological network and is believed to be more challenging to recover

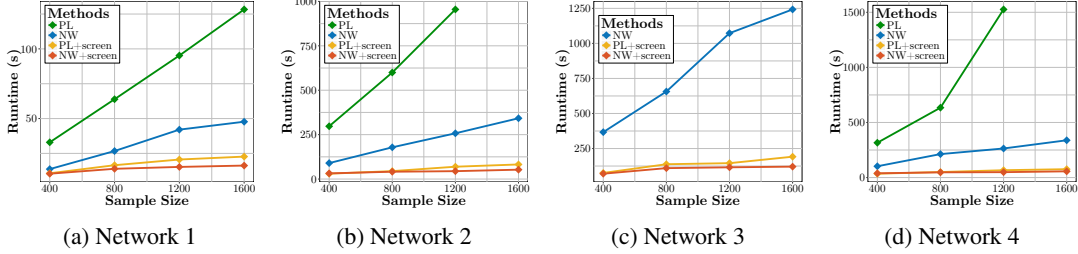

|                | (a) Network 1 | (b) Network 2 | (c) Network 3 | (d) Network 4 |

Figure 1: Runtime of pathwise optimization on networks in Table 2. Runtime plotted is the median runtime over five trials. The experiments of the baseline method PL without screening can not be fully conducted on larger networks due to high memory cost. **NW**: Node-wise logistic regression without screening; **NW+screen**: Node-wise logistic regression with screening; **PL**: Pseudolikelihood without screening; **PL+screen**: Pseudolikelihood with screening.

compared with other less complicated structures [Chen and Sharp, 2004, Peng et al., 2009, Danaher et al., 2014]. Each edge of each network is associated with a weight first sampled from a standard normal distribution, and then increased or decreased by 0.2 to further deviate from zero. For each network, 1600 samples are generated via Gibbs sampling *within each subnetwork*. Experiments on exact optimization are reported in B.2.

## 5.1 Pathwise Optimization

Pathwise optimization aims to compute solutions over a range of different $\lambda$'s. Formally, we denote the set of $\lambda$'s used in (2) as $\Lambda = \{\lambda_1, \lambda_2, \cdots, \lambda_\tau\}$, and without loss of generality, we assume that $\lambda_1 < \lambda_2 < \cdots < \lambda_\tau$.

The introduction of the screening rule provides us insightful heuristics for the determination of $\Lambda$. We start by choosing a $\lambda_1$ that reflects the sparse blockwise structural assumption on the data. To achieve sparsity and avoid densely connected structures, we assume that the number of edges in the ground truth network is $O(p)$. This assumption coincides with networks generated according to a power law degree distribution and hence is a faithful representation of the prior knowledge stemming from many biological problems. As a heuristic, we relax and apply the screening rule in (4) on each of the $\binom{p}{2}$ second empirical moments and choose $\lambda_1$ such that the number of the absolute second empirical moments that are greater than $\lambda_1$ is about $p \log p$. Given a $\lambda_1$ chosen this way, one can check how many blocks $\hat{\theta}(\lambda_1)$ has by the screening rule. To encourage blockwise structures, we magnify $\lambda_1$ via $\lambda_1 \leftarrow 1.05\lambda_1$ until the current $\hat{\theta}(\lambda_1)$ has more than one block. We then choose $\lambda_\tau$ such that the number of absolute second empirical moments that are greater than $\lambda_\tau$ is about $p$. In our experiments, we use an evenly spaced $\Lambda$ with $\tau = 25$.

To estimate the networks in Table 2, we implement both NW and PL with and without screening using `glmnet` [Friedman et al., 2010] in R as a building block for logistic regression according to Ravikumar et al. 2010 and Geng et al. 2017. To generate a symmetric parameterization for NW, we set each pair of elements from $\theta^{NW}$ in symmetric positions to the element with a larger absolute value. Given $\Lambda$, we screen only at $\lambda_1$ to identify various blocks. Each block is then solved separately in a pathwise fashion under $\Lambda$ without further screening. The rationale of performing only one screening is that starting from a $\lambda_1$ chosen in the aforementioned way has provided us a sparse blockwise structure that sets a significant portion of the parameterization to zeros; further screening over larger $\lambda$'s hence does not necessarily offer more efficiency gain.

Figure 1 summarizes the runtime of pathwise optimization on the four synthetic networks in Table 2. The experiments are conducted on a PowerEdge R720 server with two Intel(R) Xeon(R) E5-2620 CPUs and 128GB RAM. As many as 24 threads can be run in parallel. For robustness, each runtime reported is the median runtime over five trials. When the sample size is less than 1600, each trial uses a subset of samples (subsamples) that are randomly drawn from the original datasets without replacement. As illustrated in Figure 1, the efficiency gain due to the screening rule is self-evident. Both NW and PL benefit substantially from the application of the screening rule. The speedup is more apparent with the increase of sample size as well as the increase of the dimension of the data. In our experiments, we observe that even with arguably the state-of-the-art implementation [Geng et al.,

| indx | #blk | #nd/blk | TL#nd |
|------|------|---------|-------|
| 1 | 20 | 20 | 400 |
| 2 | 35 | 35 | 1225 |
| 3 | 50 | 50 | 2500 |
| 4 | 46 | 5-50 | 1265 |

Table 2: Summary of the four synthetic networks used in the experiments. `indx` represents the index of each network. `#blk` represents the number of blocks each network has. `#nd/blk` represents the number of nodes each block has. `TL#nd` represents the total number of nodes each network has.

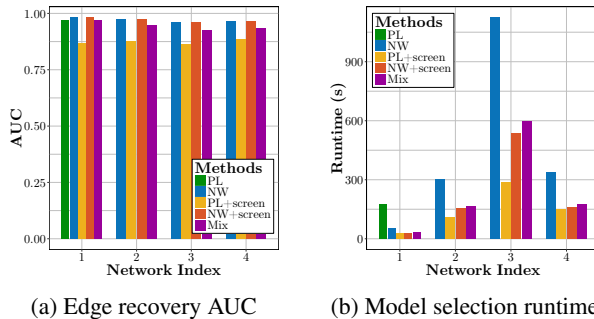

(a) Edge recovery AUC  (b) Model selection runtime

Figure 2: Model selection performance. **Mix**: provide PL+screen with the regularization parameter chosen by the model selection of NW+screen. Other legend labels are the same as in Figure 1.

2017], PL without screening still has a significantly larger memory footprint compared with that of NW. Therefore, the experiments for PL without screening are not fully conducted in Figure 1b,1c, and 1d for networks with thousands of nodes. On the contrary, PL with the screening rule has a comparable memory footprint with that of NW. Furthermore, as shown in Figure 1, after applying the screening rule, PL also has a similar runtime with NW. This phenomenon demonstrates the utility of the screening rule for effectively reducing the memory footprint of PL, making PL readily available for large-scale problems.

## 5.2  Model Selection

Our next experiment performs model selection by choosing an appropriate $\lambda$ from the regularization parameter set $\Lambda$. We leverage the Stability Approach to Regularization Selection (StARS, Liu et al. 2010) for this task. In a nutshell, StARS learns a set of various models, denoted as $\mathcal{M}$, over $\Lambda$ using many subsamples that are drawn randomly from the original dataset without replacement. It then picks a $\lambda^* \in \Lambda$ that strikes the best balance between network sparsity and edge selection stability among the models in $\mathcal{M}$. After the determination of $\lambda^*$, it is used on the entire original dataset to learn a model with which we compare the ground truth model and calculate its support recovery Area Under Curve (AUC). Implementation details of model selection are provided in B.1.

In Figure 2, we summarize the experimental results of model selection, where 24 subsamples are used for pathwise optimization in parallel to construct $\mathcal{M}$. In Figure 2a, NW with and without screening achieve the same high AUC values over all four networks, while the application of the screening rule to NW provides roughly a 2x speedup, according to Figure 2b. The same AUC value shared by the two variants of NW is due to the same $\lambda^*$ chosen by the model selection procedure. Even more importantly, it is also because that under the same $\lambda^*$, the screening rule is able to *perfectly* identify the blockwise structure of the parameterization.

Due to high memory cost, the model selection for PL without screening (green bars in Figure 2) is omitted in some networks. To control the memory footprint, the model selection for PL with screening (golden bars in Figure 2) also needs to be carried out meticulously by avoiding small $\lambda$'s in $\Lambda$ that correspond to dense structures in $\mathcal{M}$ during estimation from subsamples. While avoiding dense structures makes PL with screening the fastest among all (Figure 2b), it comes at the cost of delivering the least accurate (though still reasonably effective) support recovery performance (Figure 2a). To improve the accuracy of this approach, we also leverage the connection between NW and PL by substituting $2\lambda_{\text{NW}}^*$ for the resultant regularization parameter from model selection of PL, where $\lambda_{\text{NW}}^*$ is the regularization parameter selected for NW. This strategy results in better performance in support recovery (purple bars in Figure 2a).

## 5.3  Real World Data

Our real world data experiment applies NW with and without screening to a real world gene mutation dataset collected from 178 lung squamous cell carcinoma samples [Weinstein et al., 2013]. Each sample contains 13,665 binary variables representing the mutation statuses of various genes. For ease

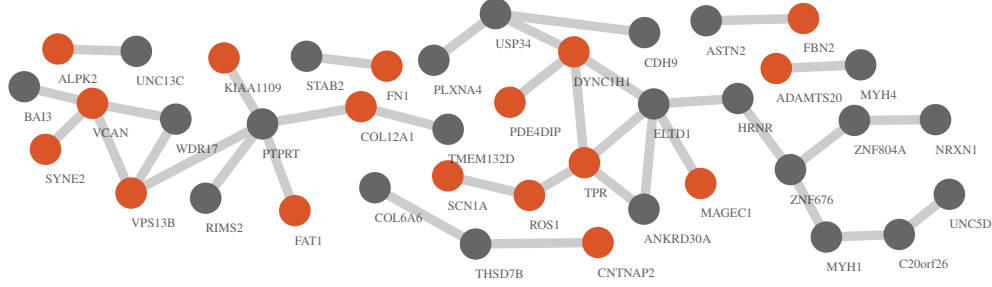

Figure 3: Connected components learned from lung squamous cell carcinoma mutation data. Genes in red are (lung) cancer and other disease related genes [Uhlén et al., 2015]. Mutation data are extracted via the `TCGA2STAT` package [Wan et al., 2015] in `R` and the figure is rendered by `Cytoscape`.

of interpretation, we keep genes whose mutation rates are at least $10\%$ across all samples, yielding a subset of 145 genes in total. We use the model selection procedure introduced in Section 5.2 to determine a $\lambda^*_{\mathrm{NW}}$ with which we learn the gene mutation network whose connected components are shown in Figure 3. For model selection, other than the configuration in B.1, we choose $\tau = 25$. 384 trials are run in parallel using all 24 threads. We also choose $\lambda_1$ such that about $2p\log(p)$ absolute second empirical moments are greater than $\lambda_1$. We choose $\lambda_\tau$ such that about $0.25p$ absolute second empirical moments are greater than $\lambda_\tau$.

In our experiment, NW with and without screening select the same $\lambda^*_{\mathrm{NW}}$, and generate the same network. Since the dataset in question has a lower dimension and a smaller sample size compared with the synthetic data, NW without screening is adequately efficient. Nonetheless, with screening NW is still roughly $20\%$ faster. This phenomenon once again indicates that in practice the screening rule can perfectly identify the blockwise sparsity pattern in the parameterization and deliver a significant efficiency gain. The genes in red in Figure 3 represent (lung) cancer and other disease related genes, which are scattered across the seven subnetworks discovered by the algorithm. In our experiment, we also notice that *all* the weights on the edges are positive. This is consistent with the biological belief that associated genes tend to *mutate together* to cause cancer.

# 6 Generalization

With unary potentials, the $\ell_1$-regularized MLE for the Ising model is defined as:

$$\hat{\theta} = \arg\min_{\theta} -\frac{1}{n}\sum_{k=1}^{n}\left(\sum_{i=1}^{p}\theta_{ii}x_i^{(k)} + \sum_{i=1}^{p-1}\sum_{j>i}^{p}\theta_{ij}x_i^{(k)}x_j^{(k)}\right) + A(\theta) + \frac{\lambda}{2}\|\theta\|_{1,\mathrm{off}}, \qquad (7)$$

where $\|\theta\|_{1,\mathrm{off}} = \sum_{i=1}^{p}\sum_{j\neq i}^{p}|\theta_{ij}|$. Note that the unary potentials are not penalized, which is a common practice [Wainwright et al., 2006, Höfling and Tibshirani, 2009, Ravikumar et al., 2010, Viallon et al., 2014] to ensure a hierarchical parameterization. The screening rule here is to replace (4) in Theorem 3 with:

$$|\mathbb{E}_{\mathbb{X}}X_iX_j - \mathbb{E}_{\mathbb{X}}X_i\mathbb{E}_{\mathbb{X}}X_j| \leq \lambda. \qquad (8)$$

Exhaustive justification, interpretation, and experiments are provided in Supplement C.

# 7 Conclusion

We have proposed a screening rule for $\ell_1$-regularized Ising model estimation. The simple closed-form screening rule is a necessary and sufficient condition for exact blockwise structural identification. Experimental results suggest that the proposed screening rule can provide drastic speedups for learning when combined with various optimization algorithms. Future directions include deriving screening rules for more general undirected graphical models [Liu et al., 2012, 2014b,a, Liu, 2014, Liu et al., 2016], and deriving screening rules for other inexact optimization algorithms [Liu and Page, 2013]. Further theoretical justifications regarding the conditions upon which the screening rule can be combined with inexact algorithms to recover block structures losslessly are also desirable.

**Acknowledgment**: The authors would like to gratefully acknowledge the NIH BD2K Initiative grant U54 AI117924 and the NIGMS grant 2RO1 GM097618.

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
