[Supplementary Material]

# Supplement

## A   Auxiliary Results

### A.1   A Lemma and a Theorem

We first show that the following lemma is true with classic graphical model inference techniques [Koller and Friedman, 2009]:

**Lemma 1.** Let $\theta \in \Theta$ be given, and let $C_l$ and $C_{l'}$ be two elements of a partition of $V$, where $l \neq l'$. If the nodes in $C_l$ are not connected with the nodes in $C_{l'}$, i.e., $\forall i \in C_l$ and $\forall j \in C_{l'}$, $\theta_{ij} = 0$, then

$$\mathbb{E}_\theta X_i X_j = \sum_{x \in \mathcal{X}} x_i x_j \mathrm{P}_\theta(x) = 0. \tag{9}$$

*Proof.* Without loss of generality, suppose $V$ is partitioned as $\{C_l, C_{l'}\}$. Since $C_l$ and $C_{l'}$ are disconnected, $\mathrm{P}_\theta(x) = \mathrm{P}_{C_l}(x)\mathrm{P}_{C_{l'}}(x)$, where $\mathrm{P}_{C_l}(x)$ and $\mathrm{P}_{C_{l'}}(x)$ represent the marginal distributions among the variables indexed by $C_l$ and $C_{l'}$, respectively. Therefore, $\forall i \in C_l$ and $\forall j \in C_{l'}$,

$$\mathbb{E}_\theta X_i X_j = \sum_{x \in \mathcal{X}} x_i x_j \mathrm{P}_\theta(x) = \sum_{x \in \mathcal{X}} x_i x_j \mathrm{P}_{C_l}(x)\mathrm{P}_{C_{l'}}(x) = \sum_{\substack{x_i, x_j \in \\ \{-1,1\}}} x_i x_j \mathrm{P}(x_i)\mathrm{P}(x_j), \tag{10}$$

By a symmetric argument, one can show that $\mathrm{P}(x_i) = \frac{1}{2}$, $\forall i \in V$. Therefore, in (10), $\mathbb{E}_\theta X_i X_j = 0$. $\qquad\square$

In (9), $\mathbb{E}_\theta X_i X_j$ represents the element at the $i^{th}$ row and the $j^{th}$ column of the expectation of the second moment of the random vector $X$ about the origin under $\mathrm{P}_\theta(x)$, $\mathbb{E}_\theta X X^\top$. The theorem establishes the sparsity pattern correspondence between $\theta$ and $\mathbb{E}_\theta X X^\top$ for *any* given $\theta \in \Theta$. In Section 3, we will see its significant role played in the derivation of the screening rule.

If we can identify the blockwise structure of $\hat{\theta}$ in advance, we can solve each block independently due to the following theorem.

**Theorem 6.** If $\hat{\theta}$ is blockwise as shown in (3), we can identify $\hat{\theta}$ by solving, $\forall l \in \{1, 2, \cdots, L\}$, separately for:

$$\hat{\theta}_l = \arg\min_{\theta_l} -\frac{1}{n} \sum_{k=1}^{n} \sum_{i=1}^{|C_l|-1} \sum_{j>i}^{|C_l|} \theta_{lij} x_i^{(k)} x_j^{(k)} + A(\theta_l) + \frac{\lambda}{2} \|\theta_l\|_1,$$

where $|C_l|$ represents the cardinality of $C_l$.

*Proof.* Theorem 6 can be proved by inspection. $\qquad\square$

### A.2   Optimality Conditions

Another essential element for the derivation of the screening rule is the Karush-Kuhn-Tucker (KKT) conditions for the $\ell_1$-regularized Ising model. Let $i \in V$, and $j > i$ be given, the KKT condition with respect to $\hat{\theta}_{ij}$ is given by:

$$\mathbb{E}_{\hat{\theta}} X_i X_j - \mathbb{E}_{\mathbb{X}} X_i X_j + \lambda t_{ij} = 0, \tag{11}$$

where $\mathbb{E}_{\mathbb{X}} X_i X_j = \frac{1}{n} \sum_{k=1}^{n} x_i^{(k)} x_j^{(k)}$'s are second empirical moments from the second empirical moment matrix $\mathbb{E}_{\mathbb{X}} X X^\top$, and $t_{ij}$ is the component of a subgradient that corresponds to $\hat{\theta}_{ij}$, with $t_{ij} = 1$ when $\hat{\theta}_{ij} > 0$, $t_{ij} = -1$ when $\hat{\theta}_{ij} < 0$, and $t_{ij} \in [-1, 1]$ when $\hat{\theta}_{ij} = 0$. Since the minimization problem for the $\ell_1$-regularized Ising model in (2) is a convex problem, the KKT conditions can be satisfied if and only if (2) reaches its optimal solution $\hat{\theta}$.

### A.3 Proof of Theorem 1

*Proof.* The rationale behind our proof is similar to that in Witten et al. 2011:

- We first prove necessity. Since $\hat{\theta}$ is blockwise, by Lemma 1, $\mathbb{E}_{\hat{\theta}} X_i X_j = 0$, for all $l$ and $l' \in \{1, 2, \cdots, L\}$, where $l \neq l'$, and for all $i \in C_l$, $j \in C_{l'}$. By the KKT condition in (11), $\lambda t_{ij} = \mathbb{E}_{\mathbb{X}} X_i X_j - \mathbb{E}_{\hat{\theta}} X_i X_j = \mathbb{E}_{\mathbb{X}} X_i X_j \Rightarrow |\mathbb{E}_{\mathbb{X}} X_i X_j| \leq \lambda$, for all $l$ and $l' \in \{1, 2, \cdots, L\}$, where $l \neq l'$, and for all $i \in C_l$, $j \in C_{l'}$. Note that we have used the fact that $\hat{\theta}_{ij} = 0 \Rightarrow |t_{ij}| \leq 1$.

- We then prove sufficiency via construction techniques. Specifically, we construct a blockwise $\tilde{\theta}$ and show that $\tilde{\theta}$ satisfies KKT conditions so that $\tilde{\theta}$ is, in fact, optimal, i.e., $\tilde{\theta} = \hat{\theta}$. For this purpose, we first set all the off-block-diagonal elements in $\tilde{\theta}$ that satisfy (4) to zeros. In this way, $\tilde{\theta}$ is blockwise with respect to the partition $\{C_1, C_2, \cdots, C_L\}$ and hence Lemma 1 can be applied. The consequence is that $\mathbb{E}_{\tilde{\theta}} X_i X_j = 0$, for all $l$ and $l' \in \{1, 2, \cdots, L\}$, where $l \neq l'$, and for all $i \in C_l$, $j \in C_{l'}$. Therefore, the KKT conditions for these off-block-diagonal zero elements of $\tilde{\theta}$ can be satisfied. Furthermore, now that $\tilde{\theta}$ is blockwise, the block diagonal elements can also be computed via exact optimization separately. In this way, the KKT conditions for the block diagonal elements of $\tilde{\theta}$ can also be satisfied. We have shown that all the elements in $\tilde{\theta}$ satisfy KKT conditions. Therefore, $\tilde{\theta}$ constructed in this way is indeed optimal and hence $\tilde{\theta} = \hat{\theta}$. $\square$

### A.4 Proof of Theorem 2

*Proof.* When $\hat{\theta} = 0$, all the nodes are disconnected from each other, which is equivalent to considering the fully disconnected partition $\{\{1\}, \{2\}, \cdots, \{p\}\}$. Using this partition, by Theorem 1, it is necessary and sufficient for $\lambda_{\max} = \max_{i,j \in V, i \neq j} |\mathbb{E}_{\mathbb{X}} X_i X_j|$ to guarantee that $\hat{\theta} = 0$. Furthermore, since $X_i, X_j \in \{-1, 1\}, \forall i, j \in V$, we have $\max_{i,j \in V, i \neq j} |\mathbb{E}_{\mathbb{X}} X_i X_j| \leq 1 \Rightarrow \lambda_{\max} \leq 1$. $\square$

### A.5 Proof of Corollary 1

*Proof.* Applying Theorem 1 to any partition with an element $\{i\}$ yields the result. $\square$

### A.6 A Toy Example

We consider a dataset with three variables and five samples. i.e. $p = 3$, and $n = 5$. Specifically,

$$
\mathbb{X} = \begin{bmatrix} -1 & 1 & -1 \\ -1 & -1 & -1 \\ -1 & -1 & -1 \\ -1 & -1 & 1 \\ 1 & -1 & 1 \end{bmatrix}, \quad \mathbb{E}_{\mathbb{X}} X X^\top = \begin{bmatrix} 1 & 0.2 & 0.6 \\ 0.2 & 1 & -0.2 \\ 0.6 & -0.2 & 1 \end{bmatrix}.
$$

Therefore, according to the screening rule (Theroem 1 or Corollary 1), if we set $\lambda = 0.2$, $X_2$ should be disconnected from $X_1$ and $X_3$ in $\hat{\theta}$. Solving the exact problem with $\lambda = 0.2$ confirms this proposition:

$$
\hat{\theta} = \begin{bmatrix} 0 & 0 & 0.4237578 \\ 0 & 0 & 0 \\ 0.4237578 & 0 & 0 \end{bmatrix}.
$$

Furthermore, with $\lambda = 0.2$,

$$
\hat{\theta}^{\mathrm{NW}} = \begin{bmatrix} 0 & 0.1013663 & 0.4479399 \\ 0 & 0 & 0 \\ 0.4479399 & -0.1013663 & 0 \end{bmatrix},
$$

$$
\hat{\theta}^{\mathrm{NW}}_{\min} = \begin{bmatrix} 0 & 0 & 0.4479399 \\ 0 & 0 & 0 \\ 0.4479399 & 0 & 0 \end{bmatrix},
$$

$$
\hat{\theta}^{\mathrm{PL}} = \begin{bmatrix} 0 & 0.06702585 & 0.43879982 \\ 0.06702585 & 0 & -0.06702585 \\ 0.43879982 & -0.06702585 & 0 \end{bmatrix}.
$$

This suggests that $X_1$, $X_2$, and $X_3$ are connected in $\hat{\theta}^{\mathrm{NW}}$ and $\hat{\theta}^{\mathrm{PL}}$, and the screening rule makes mistakes in this example. However, in $\hat{\theta}^{\mathrm{NW}}_{\min}$, $X_2$ is fully disconnected from $X_1$ and $X_3$, which is guaranteed by Theorem 5.

### A.7 Proof of Theorem 3

*Proof.* Let $j \in \{1, 2, \cdots, p-1\}$ be given, by the KKT conditions of (5), for the $\hat{\theta}^{\mathrm{NW}}_{\backslash i, j}$ component,

$$\frac{1}{n} \sum_{k=1}^{n} 2x^{(k)}_{\backslash i, j} \left( y^{(k)}_i - \frac{1}{1 + \exp\left(-\hat{\eta}^{(k)}_{\backslash i}\right)} \right) = \lambda t_j, \tag{12}$$

where $t_j$ is the $j^{th}$ component of the subgradient. Since $\lambda = \lambda^{\mathrm{NW}}_{\max} \Leftrightarrow \hat{\theta}^{\mathrm{NW}} = 0 \Rightarrow \hat{\eta}^{(k)}_{\backslash i} = 0, \forall i \in V$, $\forall k$, we have that

$$y^{(k)}_i - \frac{1}{1 + \exp\left(-\hat{\eta}^{(k)}_{\backslash i}\right)} = y^{(k)}_i - \frac{1}{2} = \frac{1}{2} x^{(k)}_i. \tag{13}$$

Substitute (13) into (12) yields $|\mathbb{E}_{\mathbb{X}} X_i X_j| \leq \lambda^{\mathrm{NW}}_{\max} = \lambda_{\max}$, where we have used the fact that $|t_j| \leq 1$ and Theorem 2. $\square$

### A.8 Proof of Theorem 4

*Proof.* We follow an argument that is similar to the proof of Theorem 3. Specifically, without loss of generality, we consider the case where $i < j$. When $\lambda = \lambda^{\mathrm{PL}}_{\max}$, by the KKT conditions of (6) with respect to $\hat{\theta}^{\mathrm{PL}}_{ij}$:

$$\left| \frac{1}{n} \sum_{k=1}^{n} \left[ 2x^{(k)}_j \left( y^{(k)}_i - \frac{1}{2} \right) + 2x^{(k)}_i \left( y^{(k)}_j - \frac{1}{2} \right) \right] \right| = \left| \frac{2}{n} \sum_{k=1}^{n} x^{(k)}_i x^{(k)}_j \right| \leq \lambda^{\mathrm{PL}}_{\max} \Rightarrow |\mathbb{E}_{\mathbb{X}} X_i X_j| \leq \frac{\lambda^{\mathrm{PL}}_{\max}}{2}.$$

Using Theorem 2 we have that $\lambda^{\mathrm{PL}}_{\max} = 2\lambda_{\max}$. $\square$

### A.9 Proof of Theorem 5

*Proof.* We first prove necessity. $\hat{\theta}^{\mathrm{NW}}_{\backslash i} = 0 \Rightarrow \hat{\eta}^{(k)}_{\backslash i} = 0, \forall k \Rightarrow$ (13) can be satisfied $\Rightarrow$ (12) can be satisfied using (13) $\Rightarrow \lambda^{\mathrm{NW}} \geq \max_{j \in V \backslash \{i\}} |\mathbb{E}_{\mathbb{X}} X_i X_j|$. Note that $\hat{\theta}^{\mathrm{NW}}_{\backslash i} = 0$ implies that $X_i$ is fully disconnected in $\hat{\theta}^{\mathrm{NW}}_{\min}$. We then prove sufficiency. To this end, $\forall j \in V \backslash \{i\}$, we set $\tilde{\theta}^{\mathrm{NW}}_{ij} = 0$. That is to say, $\tilde{\theta}^{\mathrm{NW}}_{\backslash i} = 0$. Following the same rationale behind the proof of necessity, and using the assumption that $\lambda^{\mathrm{NW}} \geq \max_{j \in V \backslash \{i\}} |\mathbb{E}_{\mathbb{X}} X_i X_j|$, the KKT conditions for $\tilde{\theta}^{\mathrm{NW}}_{\backslash i} = 0$ can be satisfied. The KKT conditions for $\tilde{\theta}^{\mathrm{NW}}_{\backslash j}$'s, where $j \in V \backslash \{i\}$ can be trivially satisfied by solving the corresponding penalized logistic regression problems. Therefore, $\tilde{\theta}^{\mathrm{NW}}$ is indeed optimal. i.e. $\tilde{\theta}^{\mathrm{NW}} = \hat{\theta}^{\mathrm{NW}}$. Furthermore, by the definition of $\hat{\theta}^{\mathrm{NW}}_{\min}$, $\left(\hat{\theta}^{\mathrm{NW}}_{\min}\right)_{ij} = \left(\hat{\theta}^{\mathrm{NW}}_{\min}\right)_{ji} = 0$ because $\tilde{\theta}^{\mathrm{NW}}_{\backslash i} = 0$. Therefore, $X_i$ is fully disconnected from the remaining nodes in $\hat{\theta}^{\mathrm{NW}}_{\min}$. $\square$

## B Experiments

### B.1 Model Selection Experiment

Our model selection procedure is a variant of that in Liu et al. 2010. To introduce enough variation, we neglect edges that do not show up in the solutions at least once under any $\lambda \in \Lambda$ when computing the total instability defined in Liu et al. 2010. We choose $\beta = 0.1$ defined in the paper. We refer interested readers to the paper for the details of StARS.

## B.2 Exact Optimization

To demonstrate the efficiency gain provided by the screening rule in exact optimization, we consider a dataset of 1600 samples generated from a network with 16 power law degree distributed subnetworks of 16 nodes. We select $\lambda_{\mathrm{NW}}^*$ using the model selection procedure in Section 5.2 and compute the exact solution under $\lambda_{\mathrm{NW}}^*$ using the proximal gradient method with constant step length [Pena and Tibshirani, 2016]. Under $\lambda_{\mathrm{NW}}^*$, the network can be successfully divided into 16 blocks according to the screening rule. Without further assumption on the structure of the subnetworks, we then compute the solution to each block separately in parallel using the NW solution as initialization. The problem can be solved in about 90 seconds. Since there are 256 nodes in the network, exact optimization in this fashion would be unimaginable had the screening rule not been applied to this problem.

## C  Generalization

Formally, the generalized screening rule for Ising models with unary potentials is given by Theorem 7.

**Theorem 7.** Let a partition of V, $\{C_1, C_2, \cdots, C_L\}$, be given. Let the dataset $\mathbb{X} = \{x^{(1)}, x^{(2)}, \cdots, x^{(n)}\}$ be given. Define $\mathbb{E}_{\mathbb{X}} X_i X_j = \frac{1}{n}\sum_{k=1}^n x_i^{(k)} x_j^{(k)}$, and $\mathbb{E}_{\mathbb{X}} X_i = \frac{1}{n}\sum_{k=1}^n x_i^{(k)}$. A necessary and sufficient condition for $\hat{\theta}$ to be blockwise with respect to the given partition is that

$$|\mathbb{E}_{\mathbb{X}} X_i X_j - \mathbb{E}_{\mathbb{X}} X_i \mathbb{E}_{\mathbb{X}} X_j| \leq \lambda,$$

for all $l$ and $l' \in \{1, 2, \cdots, L\}$, where $l \neq l'$, and for all $i \in C_l, j \in C_{l'}$.

### C.1  A Lemma

To show that Theorem 7 is true, we first show that the following lemma is true:

**Lemma 2.** Let $\theta$ be given, and let $C_l$ and $C_{l'}$ be two elements of a partition of $V$, where $l \neq l'$. If the nodes in $C_l$ are not connected with the nodes in $C_{l'}$, i.e., $\forall i \in C_l$ and $\forall j \in C_{l'}$, $\theta_{ij} = 0$, then

$$\mathbb{E}_\theta X_i X_j = \mathbb{E}_\theta X_i \mathbb{E}_\theta X_j. \tag{14}$$

*Proof.* Without loss of generality, suppose $V$ is partitioned as $\{C_l, C_{l'}\}$. Since $C_l$ and $C_{l'}$ are disconnected, $\mathrm{P}_\theta(x) = \mathrm{P}_{C_l}(x)\mathrm{P}_{C_{l'}}(x)$, where $\mathrm{P}_{C_l}(x)$ and $\mathrm{P}_{C_{l'}}(x)$ represent the marginal distributions among the variables indexed by $C_l$ and $C_{l'}$, respectively. Therefore, $\forall i \in C_l$ and $\forall j \in C_{l'}$,

$$\mathbb{E}_\theta X_i X_j = \sum_{x \in \mathcal{X}} x_i x_j \mathrm{P}_\theta(x) = \sum_{x \in \mathcal{X}} x_i x_j \mathrm{P}_{C_l}(x)\mathrm{P}_{C_{l'}}(x)$$

$$= \sum_{\substack{x_i, x_j \in \\ \{-1,1\}}} x_i x_j \mathrm{P}(x_i)\mathrm{P}(x_j) = \left(\sum_{x_i \in \{-1,1\}} x_i \mathrm{P}(x_i)\right)\left(\sum_{x_j \in \{-1,1\}} x_j \mathrm{P}(x_j)\right)$$

$$= \mathbb{E}_\theta X_i \mathbb{E}_\theta X_j.$$

$\square$

### C.2  Optimality Conditions

Consider the KKT conditions for (7). The KKT condition for $\hat{\theta}_{ii}$ is:

$$\mathbb{E}_{\mathbb{X}} X_i = \mathbb{E}_{\hat{\theta}} X_i. \tag{15}$$

The KKT condition for $\hat{\theta}_{ij}$, where $i \neq j$, is:

$$\mathbb{E}_{\hat{\theta}} X_i X_j - \mathbb{E}_{\mathbb{X}} X_i X_j + \lambda t_{ij} = 0. \tag{16}$$

### C.3  Proof of Theorem 7

*Proof.* We first prove necessity. Since $\hat{\theta}$ is blockwise, by Lemma 2, $\mathbb{E}_{\hat{\theta}} X_i X_j = \mathbb{E}_{\hat{\theta}} X_i \mathbb{E}_{\hat{\theta}} X_j$, for all $l$ and $l' \in \{1, 2, \cdots, L\}$, where $l \neq l'$, and for all $i \in C_l, j \in C_{l'}$. By the KKT condition in

(15) and (16), $\lambda t_{ij} = \mathbb{E}_{\mathbb{X}} X_i X_j - \mathbb{E}_{\hat{\theta}} X_i X_j = \mathbb{E}_{\mathbb{X}} X_i X_j - \mathbb{E}_{\hat{\theta}} X_i \mathbb{E}_{\hat{\theta}} X_j = \mathbb{E}_{\mathbb{X}} X_i X_j - \mathbb{E}_{\mathbb{X}} X_i \mathbb{E}_{\mathbb{X}} X_j \Rightarrow$
$|\mathbb{E}_{\mathbb{X}} X_i X_j - \mathbb{E}_{\mathbb{X}} X_i \mathbb{E}_{\mathbb{X}} X_j| \le \lambda$, for all $l$ and $l' \in \{1, 2, \cdots, L\}$, where $l \ne l'$, and for all $i \in C_l$, $j \in C_{l'}$. Note that we have used the fact that $\hat{\theta}_{ij} = 0 \Rightarrow |t_{ij}| \le 1$.

We then prove sufficiency via construction techniques. Specifically, we construct a blockwise $\tilde{\theta}$ and show that $\tilde{\theta}$ satisfies KKT conditions so that $\tilde{\theta}$ is, in fact, optimal, i.e., $\tilde{\theta} = \hat{\theta}$. For this purpose, we first set all the off-block-diagonal elements in $\tilde{\theta}$ that satisfy (4) to zeros. In this way, $\tilde{\theta}$ is blockwise with respect to the partition $\{C_1, C_2, \cdots, C_L\}$ and hence Lemma 2 can be applied. The consequence is that $\mathbb{E}_{\tilde{\theta}} X_i X_j = \mathbb{E}_{\tilde{\theta}} X_i \mathbb{E}_{\tilde{\theta}} X_j$, for all $l$ and $l' \in \{1, 2, \cdots, L\}$, where $l \ne l'$, and for all $i \in C_l$, $j \in C_{l'}$. Therefore, the KKT conditions for these off-block-diagonal zero elements of $\tilde{\theta}$ can be satisfied. Furthermore, now that $\tilde{\theta}$ is blockwise, the block diagonal elements can also be computed via exact optimization separately. In this way, the KKT conditions for the block diagonal elements of $\tilde{\theta}$ can also be satisfied. We have shown that all the elements in $\tilde{\theta}$ satisfy KKT conditions. Therefore, $\tilde{\theta}$ constructed in this way is indeed optimal and hence $\tilde{\theta} = \hat{\theta}$. $\qquad\square$

### C.4 Interpretations

A most noteworthy consequence of Theorem 7 is that the blockwise structure of an Ising model with unary potentials can be identified *in the exact same way* as the blockwise structure of a Gaussian graphical model. This can be seen by comparing Theorem 7 with the results in Witten et al. [2011], and Mazumder and Hastie [2012]. Such a correspondence between Ising models and Gaussian graphical models have striking implications.

Since Gaussian graphical models enjoy the precious property that the sparsity pattern of its precision matrix corresponds to the sparsity pattern of its structure, it might not be surprising that a screening rule for sample covariance matrix can offer an effective approach to identify the blockwise structure of a Gaussian graphical model. On the contrary, in the regime of Ising models, in general there is no element-to-element exact sparsity pattern equivalence. Nonetheless, granted by Theorem 7, the block structure of an Ising model with unary potentials can still be identified by the same procedure as in the Gaussian case, which establishes an easily *verifiable* correspondence between the sample covariance matrix and the underlying structure for Ising models. This verifiable correspondence also distinguishes our work from Loh et al. [2012, 2013], where the correspondence between an *unverifiable* generalized precision matrix and the structure of a discrete graphical model is established. Our work is also different from Loh et al. [2012, 2013] in terms of the objective functions. While we consider the optimization perspective of the MLE problem in this work, the log-determinant problem is considered in Loh et al. [2012, 2013] with an emphasis on statistical consistency.

Furthermore, to the best of our knowledge, the screening rule in Witten et al. [2011] and Mazumder and Hastie [2012] is the strongest safe blockwise screening for Gaussian graphical models in the literature. Given the general intractability of discrete graphical model learning via maximum likelihood, the same safe screening achieved for Ising models provides an especially valuable and desperately needed guarantee that is as strong as the best known result for its polynomial-time Gaussian counterpart.

### C.5 Experiments

To demonstrate the utility of the screening rule for Ising models with unary potentials, we generate a network that consists of 40 power law degree distributed subnetworks of 20 nodes. The weights on the edges are generated in the same way as in Section 5. The weights on all nodes are set to be $0.1$ for simplicity. As many as 1600 samples are used for learning. Figure 4 reports the runtime as well as the AUC of pathwise optimization using NW with and without screening for Ising models with unary potentials. The phenomenon we observed in this case is consistent with the phenomenon for Ising models with only pairwise potentials. The screening can accelerate learning tremendously and in this experiment even delivers lossless screening. This can be seen from Figure 4b, where the AUC v.s. $\lambda$ curves of NW with and without screening completely overlap with each other.

(a) Runtime v.s. Sample size

(b) AUC v.s. $\lambda$ using all samples

Figure 4: Runtime and support recovery performance for Ising models with unary potentials. Note that in in Figure 4b, the two curves overlap.