[Reviews · NeurIPS 2017]

Reviewer 1



The author(s) present a screening rule for block structure identification in Ising model. This screening rule combined with exact or inexact optimization procedure can lead to scalebale parameter estimation in Ising model. I have following thoughts on the proposed approach- (1) Finding a block structure can also be done by using spectral clustering or regularized spectral clustering. Once the block structure is recovered then the exact or inexact optimization can be carried out there as well. How different (or may be effective) the screening procedure compared to a eigenvalue decomposition on a graph Laplacian? (2) The second moment screening rule procedure proposed here is similar to the screening rule proposed in Witten et al. (2011) and Mazumder & Hastie (2012). How different/similar is the current approach with the existing ones in the literature? (3) How robust is the block wise parametrization of $theta$ is? If the true $\theta$ does not have a block structure is the initial screening rule could lead to inaccurate parameter estimation?

Reviewer 2



SUMMARY: ======== The authors propose an efficient screening for sparse Ising model structure estimation. The screening rule is shown to be optimal for locating non-interacting variables in an L1 regularized MLE. Combining this screening rule with approximate optimization allows efficient parallel computation of the full model structure, as demonstrated on several synthetic graph problems. PROS: ===== The screening rule can be calculated fairly efficiently, in time quadratic in the number of data elements, and as such could be easily adopted in practice. The authors also clearly show that the method is optimal in a regularized MLE sense, and computation of the screening rule also provides a range of valid regularization parameters including those that yield a fully disconnected graph. CONS: ===== The authors do not discuss the relationship between the proposed screening rule and the *nearly* identical "correlation decay" property considered in Loh and Wainwright (2013) and references therein. The work of Loh and Wainwright is referenced but not cited in the text. This oversight is curious since Loh and Wainwright provide a probabilistic bound of correct reconstruction using the nodewise method considered in the present paper. This reviewer feels that the experimental results are lacking and too much material is referenced in the supplement. In particular, the authors claim experiments on real world data (L:228). However, the real world experiment is a single gene interaction network on which only the node-wise method is performed, and is entirely contained in the supplement (B.6). Moreover, the real world experiment is a small graph for which screening only yields a 20% speed improvement. This is an unfortunate omission as more detailed experimentation of real world data would have been beneficial. In addition to the real world experiment, other experimental details are deferred to the supplement or not discussed at all. Discussion of model selection is omitted and references B.4, which in turn provides only a few sentences and references Liu et al. (2010). Approximations are only compared to exact inference in a single experiment (also in the supplement) for which exact computation is computed in 90s. Finally, inclusion of the "mixed" approach which uses NW for model selection and PL for estimation is both slower than PL and less accurate than NW, it is unclear why this approach would be considered.

Reviewer 3



Review of paper 477 "A Screening Rule for l1-Regularized Ising Model Estimation" This paper presents a simple closed-from screening for l1-regularized Ising model estimation. Specifically, the authors prove (in Theorem 1) a necessary and sufficient condition that ensures that blockwise solution of the l1-regularized maximum likelihood estimation is identical the complete l1-regularized maximum likelihood solution. This result is based on the KKT conditions for the l1-regularized MLE and is elegant both in terms of its simplicity and the simplicity of its proof. The authors also discuss the challenges remaining, i.e., that the NP-hard problem is now reduced to a set of lower dimension yet still NP-hard problems. Hence, making progress in the right direction. Overall, an elegant result worthy of publication in NIPS. A thought: 1. How does the hyper parameter tuning (finding \lambda) affect this problem? If one scans through lambda, then for low enough values of \lambda only one block can be found and the computation time will not be reduced. Some clarification would be helpful.